# Mechanism-Driven and Clinically Focused Development of Botanical Foods as Multitarget Anticancer Medicine: Collective Perspectives and Insights from Preclinical Studies, IND Applications and Early-Phase Clinical Trials

**DOI:** 10.3390/cancers15030701

**Published:** 2023-01-23

**Authors:** Xiaoqiang Wang, Yin S. Chan, Kelly Wong, Ryohei Yoshitake, David Sadava, Timothy W. Synold, Paul Frankel, Przemyslaw W. Twardowski, Clayton Lau, Shiuan Chen

**Affiliations:** 1Department of Cancer Biology & Molecular Medicine, Beckman Research Institute, City of Hope, 1500 E. Duarte Rd., Duarte, CA 91010, USA; 2Department of Medical Oncology & Therapeutics Research, City of Hope Comprehensive Cancer Center, 1500 E. Duarte Rd., Duarte, CA 91010, USA; 3Department of Computational and Quantitative Medicine, Beckman Research Institute, City of Hope, 1500 E. Duarte Rd., Duarte, CA 91010, USA; 4Department of Urologic Oncology, Saint John’s Cancer Institute, 2200 Santa Monica Blvd, Santa Monica, CA 90404, USA; 5Department of Surgery, City of Hope Comprehensive Cancer Center, 1500 E. Duarte Rd., Duarte, CA 91010, USA

**Keywords:** botanical food products, botanical drugs, cancer target therapy, preclinical study, investigational new drug, early-phase clinical trials

## Abstract

**Simple Summary:**

In this review, we have conducted an extensive literature review and have drawn from our collective experiences to outline critical insights in designing preclinical studies for botanical foods as anticancer investigational new drug (IND) applications as well as to highlight the critical criteria when initiating early-phase clinical trials for botanical food products. This review is for those who wish to understand how botanical foods and their derivatives can contribute to cancer pathobiology, as well as to develop relevant therapeutic approaches to treat cancer by conducting clinically oriented and evidence-based translational studies under United States Food and Drug Administration (USFDA) regulatory frameworks.

**Abstract:**

Cancer progression and mortality remain challenging because of current obstacles and limitations in cancer treatment. Continuous efforts are being made to explore complementary and alternative approaches to alleviate the suffering of cancer patients. Epidemiological and nutritional studies have indicated that consuming botanical foods is linked to a lower risk of cancer incidence and/or improved cancer prognosis after diagnosis. From these observations, a variety of preclinical and clinical studies have been carried out to evaluate the potential of botanical food products as anticancer medicines. Unfortunately, many investigations have been poorly designed, and encouraging preclinical results have not been translated into clinical success. Botanical products contain a wide variety of chemicals, making them more difficult to study than traditional drugs. In this review, with the consideration of the regulatory framework of the USFDA, we share our collective experiences and lessons learned from 20 years of defining anticancer foods, focusing on the critical aspects of preclinical studies that are required for an IND application, as well as the checkpoints needed for early-phase clinical trials. We recommend a developmental pipeline that is based on mechanisms and clinical considerations.

## 1. Introduction

Although extensive efforts have been dedicated to improving therapeutic strategies, cancer remains a significant global health burden and a leading cause of death [1]. For many decades, cancer treatment strategies have been limited to conventional approaches such as surgery, radiotherapy, and cytotoxic chemotherapy using single agents or combinations. In addition to surgical excision and local radiotherapy, cytotoxic chemotherapy still serves as the first-line therapy for most cancer patients but has reached a therapeutic plateau [2]. To address the gaps in cancer care, effective and evidence-based curative interventions need to be tailored for and integrated into different cancer therapeutic strategies [3].

A growing number of multi-omics technologies, as well as advanced experimental and computational tools, have allowed researchers to delve deeper into the complexities of cancer [4]. Within the last two decades, cancer discoveries have been made in terms of genetic alterations in cells and tissue biology, and responses to therapy at the molecular and cellular levels. These investigations have revealed that cancer is not a single disease [5], but a heterogeneous group of diseases, where each is characterized by phenotypic and genotypic hallmarks [6]. These new pieces of knowledge have changed treatment paradigms, most notably in two ways. First, molecular profiling and subtyping of tumors have led to precision cancer therapy, where treatments can be individualized toward subsets of cancer cells harboring druggable genetic or genomic abnormalities [7]. This concept has contributed to the development of small molecules and monoclonal antibodies that stop uncontrolled cancer cell growth by: (1) interrupting oncogenic signals that are responsible for tumor growth and cell division (e.g., anti-HER2, anti-EGFR, and CDK4/6 inhibitors); (2) initiating programmed cancer cell death (e.g., PPARi and anti-bcl2); and (3) starving cancer cells by depriving them of the hormones/growth factors they need to grow (e.g., anti-estrogens and -androgens). The second paradigm is targeting critical molecular and cellular components of the tumor immune microenvironment, such as engineered antitumor immune cells (e.g., CAR-T/CAT-NK) and immune checkpoint inhibitors (e.g., anti-PD-1/L1 and anticytotoxic T lymphocyte antigen-4 monoclonal antibodies [CTLA-4]) [8].

Although significant advances in therapy have been made for oncogene-driven cancers and in immuno-oncology, both approaches still face significant challenges because such treatments are effective only in a subset of patients whose cancers express the relevant targets [9,10]. Often, cancers are difficult to treat because their progression and response to treatment are determined by network architectures rather than a single molecular target or cell type [11].

To achieve long-term survival benefits, drug combinations simultaneously target multiple molecular alterations or cellular compartments, and thus have been extensively discussed and actively tested in preclinical and clinical settings [12,13]. The cancer treatment paradigm is shifting from the traditional “one-drug/one-target/one-disease” approach to a “network-targets/multiple-component therapeutics” approach [14,15]. Designing these multiple therapeutic approaches will be a challenging but promising cancer therapeutic strategy in the future [16,17].

The shift in the approach to cancer therapy has led to the search for multitarget anticancer drugs [18,19]. Because of their heterogeneous chemical properties, botanical medications have become a focus for potential therapies due to their ability to target multiple oncogenic signaling pathways and cellular compartments simultaneously [20,21,22]. Botanical foods such as edible plants (e.g., fruits, vegetables, nuts, and grains), edible mushrooms (e.g., white button, shiitake, maitake, and oyster), and edible algae (e.g., seaweed) contain many constituents (carotenoids, phenolic acids, polyphenols, carbohydrate polymers, and lipids) and have anticarcinogenic properties with or without low systemic toxicity [23,24]. However, much of this knowledge has come from limited research on a few cancer cell lines, and there is seldom evidence from animal models or human clinical trials [25,26]. Although many widely used drugs were originally isolated from medicinal plants [27], the US Food and Drug Administration (USFDA) has never approved any food-derived botanical product as an anticancer medicine [28]. Unlike the traditional drug development pipeline with a defined molecule, the drug development of botanical products from foods is challenging due to their chemical heterogeneity [29]. To define the utility of the natural constituents present in botanical foods and to translate affordable and targeted cancer therapy strategies to cancer patients, significant efforts are needed from mechanism-driven and clinically relevant testing and standardization.

The intent of this review is not to provide an overview of anticancer foods along with their phytochemical profile and mechanism of action, because there have been a wide variety of review articles in this direction. Rather, we would like to share our collective experiences in the characterization of anticancer foods under USFDA regulation. The Botanical Drug Development Guidance for Industry was issued by the USFDA in 2016 [30], whereas instructional and explanatory publications were published in 2019 [28] and 2020 [31]. Under the regulatory framework of the USFDA, the development of botanical foods as anticancer medicines is a stepwise procedure that involves multidisciplinary approaches. The core questions of the entire process would relate to efficacy/safety studies and regulatory issues. Here, we draw from our experiences to outline critical insights in designing preclinical studies for investigational new drug (IND) applications as well as to highlight the critical criteria when initiating early-phase clinical trials for INDs. We present a step-by-step pipeline for developing anticancer drugs from botanical foods based on a comprehensive analysis of current obstacles and progress. The steps that are unique to or different from botanical foods, i.e., those pertaining to conventional single-agent anticancer drugs, are indicated by the star-shaped icons (Figure 1).

Table 1 provides explanations and summary of seven critical checkpoints that are considered in developing botanical foods as anticancer medicine from our collective experiences. The points that are particularly in need of botanical foods are emphasized with bold characters.

## 2. From Botanical Foods to Anticancer Therapeutics: The Science beyond the Myth

The public and social media have widely promoted the concept of “anticancer super foods” when referring to foods that claim to protect humans against cancer incidence and progression. While the concept may have been exaggerated, there is some validity to the claims. “Anticancer super foods” largely consist of botanical foods, such as edible mushrooms, berries, broccoli, tomatoes, walnuts, grapes, and others [32]. As noted above, botanical foods contain phytochemicals that display cancer therapeutic potential based on their ability to interfere with key molecular events or networks in cancer initiation and progression [25,33]. A wide range of laboratory and clinical settings have been explored to investigate the anticancer properties of botanical foods. Many reviews and books have been written about the progress made [34,35,36,37], but health care professionals face a dilemma when advising their patients about consuming “anticancer foods”. Physicians and nutritionists are hesitant to offer any straightforward and actionable advice because there is a lack of evidence from scientifically conducted clinical trials [38,39,40].

To fully understand the obstacles faced by health care professionals, we have evaluated several professional sources (Table 2) to extract clinically useful or applicable information on “anticancer foods”. The Complementary and Alternative Medicine for Health Professions website by the National Cancer Institute (NCI) provides evidence-based Physician Data Query (PDQ) information on over 200 types of food and food-derived supplements, which include information about how drugs interact with food and dietary supplements during cancer therapy [41]. “About Herbs, Botanical & Other Products,” a website created by the Memorial Sloan Kettering Cancer Center, provides evidence-based information on foods and herbs, clinical summaries, uses and benefits, mechanisms of action, herb–drug interactions, and warnings [42]. “Foods and Herbs in the Natural Medicines Comprehensive Database” is an encyclopedic web resource that provides references on safety, adverse effects, dosing and administration, effectiveness, drug–herb interactions, lab interactions, pharmacokinetics, and mechanisms of action [43,44,45]. The American Cancer Research Institute, the NCI drug dictionary [46], and the United State Department of Agriculture (USDA) Food Data [47] are other sources [48] we also investigated.

Our review of these six databases has led to the identification of the ten botanical foods that have evidence-based anticancer activity (Table 3). In the table, we focus on the information from food–drug interactions in cancer therapy. A more comprehensive presentation of information regarding bioactive molecules in foods, their mechanisms of action, and functional endpoints is provided in Appendix A. Aside from the foods listed in Table 3, there are several other botanical products that are also studied for their anticancer activities, such as tea [49], coffee [50], and spinach [51]. The six databases listed in Table 2 provide comprehensive information about these botanicals.

Botanical food products are widely accepted as non-toxic agents, but consuming botanical products derived from plants may expose cancer patients to risk from food–drug interactions related to cytochrome P450 (CYPs) enzymes [97,98]. CYPs are heme monooxygenases that convert endogenous and xenobiotic substrates (drugs, phytochemicals, etc.) into metabolites that can be readily excreted or transported [99]. Hepatic and intestinal tissues are the primary sites of expression for CYPs, but expression can also be found in other organs and tissues, such as the kidney [100]. Studies have demonstrated that bioactive compounds from botanical foods (e.g., fruits, vegetables, mushrooms, and grains) can inhibit or induce the activity of CYPs, which may alter drug modification/levels and the magnitude of exposure, leading to toxicity and adverse events [97,98]. Approaches for studying food/herb–drug interactions include, but are not limited to, conventional biochemistry and enzymatic assay, cell-based studies, ex vivo, and animal studies. For example, in an earlier study on food/herb–drug interactions, we used rat liver purified NAD(P)H: quinone acceptor oxidoreductase (NQO1) to conduct enzyme kinetic assays. From there, we have determined that flavonoids isolated from Chinese herbs inhibit NQO1 in a fashion identical to that of dicoumarol, an anticoagulant drug. The study’s findings suggest that these herbs may be linked to the mechanism for blood coagulation, and thus should be used with caution by patients prescribed anticoagulant drugs [101]. From the standpoint of safety, cancer patients and health care providers should be aware of the possible interactions between plant-derived products and other prescribed drugs. 

Although some health care providers are enthusiastic about using plant foods as cancer therapy from the limited data obtained in in vitro, in vivo, and in silico approaches, most require much more knowledge of the mechanisms of action of these agents. Before designing preclinical and clinical studies, it is important to be aware of the unique physiological and pharmacological processes that take place in the human body when consuming botanical foods or related products compared to the processes involved in consuming conventional single-agent synthetic drugs. Figure 2 illustrates how botanical food products (whole foods, food extracts, and mixtures taken as dietary supplements) are liberated via oral consumption and are subsequently digested in digestive organs after oral consumption. The gastrointestinal tract, particularly the small intestine, plays a central role in processing food products, where bioavailable chemical constituents are absorbed and metabolized by intestinal epithelia [102,103]. Afterward, the bioactive molecules are transported by the bloodstream to target organs, where the bioactive constituents exert their anticancer effects once it is delivered to cancer tissues/cells 

Unlike synthetic drugs that are artificially modified for bioavailability and therapeutic effects, the bioactive compounds such as carotenes, flavonoids, and phenols found in botanical foods are naturally occurring chemicals. Therefore, the intrinsic bioavailability of functional molecules is a critical factor, which can ultimately affect their anticancer activity [104]. The physical, chemical, and biological assessments of the bioavailability of the active constituents in candidate food products are essential in the early phase of development. Botanical food products that contain orally bioavailable chemicals that have or are suspected to have anticancer properties are candidates for in-depth investigations.

In addition to bioavailable phytochemicals, extracellular vehicles (EVs) derived from plant foods are emerging as novel functional components. EVs are lipoprotein structures with a wide range of sizes, ranging from 20 to 500 nm. The fruits, flesh, and roots of many plant species can be used to isolate plant-derived EVs (P-EVs), which contain proteins, RNAs, and lipids [105]. From immunomodulation to cancer development, EVs play an influential role in human physiological and pathological processes [106]. Consisting of a lipid bilayer, EVs are stable and are orally bioavailable, and therefore, P-EVs have been proposed as promising therapeutic agents [107]. In a recently published study, broccoli-derived EVs were used to package RNA-based drugs to improve their bioavailability and stability before reaching target organs [108]. As a novel drug-delivery carrier, P-EVs hold promising potential for clinical application in the treatment of different diseases including cancers [109].

The small intestine is also the site where food-induced immune responses are processed [110]. Gut-associated lymphoid tissues (GALTs) prime the immune response by triggering and promoting systemic immune responses in the lamina propria, intraepithelial lymphocytes, mesenteric lymph nodes, Peyer’s patches, and isolated lymphoid follicles [111,112]. For instance, carbohydrate polymers and/or polysaccharides, which are enriched in edible mushrooms and whole grains, are captured by GALT, where the gut-resident immune cells are primed, the immune-activating signals (e.g., cytokines and primed innate immune cells) [113,114,115] consequently triggers anticancer immunity by provoking both innate immune cells and adaptive immune cells (neutrophils, dendritic cells, T/NK cells) [116] and/or reprograms tumor immunosuppressive cells (M2-macrophage, myeloid-derived suppressive cells [MDSCs]), ultimately leading to immune-mediated anticancer activity [117].

In addition to the gut epithelial barrier and mucosal immune system, gut-microbiome (including microbiome-encoded enzymes) has proven as another determinant of drug pharmacokinetics and therapeutic response by modifying the drugs’ physicochemical properties (i.e., solubility and permeability) and/or transforming the drugs’ activity (i.e., microbe-mediated prodrug activation, or drug metabolism or inactivation) [118]. Particularly for botanical food products which are orally consumed, extensive research has suggested a mutual interface between the gut microbiome and botanical food products. Phytochemicals present in botanical foods can modulate the gut microbiota composition by selectively inhibiting some pathogenic microorganisms, thus reducing competition within microbial populations, and ultimately promoting homeostasis within the gastrointestinal tract [119]. Furthermore, the gut microbiome-encoded enzymes metabolize phytochemicals, in turn, may also increase their bioavailability and bioactivity [120]. In brief, inter-individual variability in gut microbiome composition would influence individual response to botanical food’s anticancer response. Therefore, the researchers must not overlook the impacts on/from gut-microbiome in botanical food intervention studies.

In summary, scientists could better conceptualize and design preclinical and clinical research toward transforming botanical foods into anticancer therapies if they have a comprehensive understanding of the interface between food and drugs, as well as the physiology and pharmacology of foods in humans.

## 3. Designing the Preclinical Study: Consideration of the Regulatory, Scientific, and Clinical Settings Prior to Human Studies

Under the regulatory framework in the United States, botanical foods include edible plants, algae, fungi, and combinations thereof [30]. If a study’s ultimate goal is to investigate whether the botanical product may “*treat, cure or prevent any disease involving safety and clinical endpoint evaluation*”, the USFDA needs to approve an Investigational New Drug (IND) application prior to conducting human studies. Published by the USFDA in 2016, the *Botanical Drug Development Guidance for Industry* [30] states that IND applications must include: (1) the source, chemistry, and manufacturing process of “the candidate drug”, in addition to prior human use experience and clinically relevant biological assays to demonstrate a consistency of action; (2) adequate preclinical toxicology/pharmacology data to assess the efficacy versus the safety of the testing drug, particularly if the doses and durations are higher than in prior trials; (3) pharmacokinetics and pharmacodynamics (PK/PD) information and evaluation of potential interactions with other drugs or botanicals; and (4) the human study protocol, including human subject enrollment, dosing and administration pattern, safety evaluation criteria, and clinical endpoints. Most of the information can be determined and justified primarily through thoughtfully designed preclinical studies. To ensure a successful translation of preclinical findings into clinical settings, it is imperative to incorporate regulatory requirements, clinical needs and scientific questions when designing preclinical studies.

### 3.1. Establishment of a Mechanism-Based and Clinically Relevant In Vitro Bioassay

The traditional method of developing drugs from natural products uses defined chemical constituents, such as isolated pure natural compounds or synthetic chemicals based on natural products [121]. However, botanical foods contain various types and levels of functional molecules, and therefore it is often impractical to chemically identify and quantify all the active constituents within the botanical products [122]. A cytotoxic assay of cancer cells in culture is frequently used in preclinical studies as a demonstration of the anticancer activity of candidate compounds, including botanical drugs and food extracts. A classic example of a cytotoxic assay is the colorimetric MTT/MTS assay, which measures overall cellular metabolism via NADPH oxidoreductases by the formation of an insoluble colored formazan [123]. When the candidate drug significantly reduces the activity of the MTS/MTT test of a cancer cell line, the drug is considered to assert a “promising anticancer effect”. Unfortunately, this type of study is only based on results from a simple cytotoxic and colorimetric assay is inadequate because: (1) cytotoxicity is not a synonym for anticancer activity because cytotoxicity describes the cell-killing ability of given treatments in both normal and malignant cells [124]; and (2) colorimetric assay would yield false positive/negative data on most plant extracts contain natural pigments, particularly flavonoids, carotenoids, and chlorophylls [125]. When dealing with a botanical drug consisting of a heterogenous chemical mixture, the USFDA suggests developing a mechanism-based and clinically relevant in vitro biological assay that can reflect the product’s known or intended mechanism of action and/or relate to the surrogate diagnostic biomarker of the cancers [30].

### 3.2. Selection of Clinically Representative Preclinical Models: From Wet (In Vitro, Ex Vivo, and In Vivo) to Dry (In Silico)

Following the drug development dogma of “safety vs. efficacy”, all anticancer drugs, including botanical products, must be evaluated for therapeutic surrogate markers to determine their efficacy as well as rational endpoints to reduce treatment-related risks. Some standard criteria include, but are not limited to, disease indication, targeted population selection, pharmacology/toxicology, a biomarker system to predict the effect, and a way to monitor the adverse events [126]. These standards can be implemented in the appropriate preclinical models, which can increase the success of clinical trials and eventually, a productive anticancer drug development [127]. Unfortunately, there is a significant in vitro/in vivo disconnection in botanical product anticancer medicine development due to the complexity of chemical profiles in botanical foods when compared to single small molecules with defined physical, chemical, and biological properties [128]. 

However, the development of sophisticated clinically relevant ex vivo models, such as organs on a chip (OoCs) with human microsomal systems and patient-derived organoids, may assist in drug development with botanical food molecules and in evaluating between drug indications with an accurate recreation of the human organ function [129]. More importantly, combining the multi-omics-based in silico model, network pharmacology, and single-cell and spatial multi-omics techniques can transform the traditional “one-drug/one-target/one-disease” approach into a new approach that implements “network-targets, multiple-component-therapeutics” [130]. These new and advanced approaches will provide critical insight into current trials and inspire further improvements.

#### 3.2.1. In Vitro Models: Choose and Use the Cancer Cell Lines Wisely

Cancer cell lines are straightforward in vitro models to use in drug development [131]. However, it is important to consider that cancer cell lines have their limitations and drawbacks. Some of these difficulties include: (1) single-cell lines culture in a two-dimensional (2D) dish do not interact with other cell types typically present in the tumor microenvironment; (2) the original tissue architecture is lost, thus the growth of the cancer cells is not influenced by paracrine chemicals [132]; and (3) it is difficult to replicate the effects of in vivo drug absorption, distribution, metabolism, and excretion (ADME) in cancer cell lines [133]. These limitations suggest that cancer cell lines in culture may not accurately address the critical mechanisms that influence a drug’s action in vivo, but due to their endless supply of biological materials for experiments, they still provide the largest variety of in vitro models for obtaining mechanistic and therapeutic insights [132]. Choosing cancer cell lines for single-agent drugs with defined targets and/or mechanisms of action is straightforward. Cancer cell lines that have matched the histological and molecular features of their respective cancers should be considered in vitro model systems of the diseases from which they are derived [134,135]. For instance, the American Type Culture Collection (ATCC) provides 46 cancer cell panels that are grouped by either tissue of tumor origin (e.g., breast cancer cell panel and colon cancer panel,) or by the molecular signature (e.g., EGFR genetic alteration cell panel and PI3K genetic alteration cell panel). For botanical products with a presumed mechanism of action or treatment indications, the concept that pertains to conventional anticancer drugs would be also suitable.

However, for botanical products with unclear mechanisms or multitarget potentials, it is crucial to use multiple cancer cell lines along with normal cell lines at the early stages of drug screening to determine the treatment indications and mechanism of action [135]. The screening strategy set up by the US NCI on the 60-cancer-cell-line panel (NCI-60) should be considered [136]. In studies where NCI-60 was used for botanical product screening, researchers have characterized the potential mechanism of natural products by examining associations of molecular genomic features in NCI-60 cancer cell line panels with in vitro response by treating with compounds derived from botanical natural products. From there, the author has suggested potential mechanisms of action of certain natural products and their novel associations of in vitro response with gene expression [137].

#### 3.2.2. Ex Vivo Models: Patient-Derived Organoids and Organs on a Chip

As discussed previously, the drawback of cell line models for anticancer drug development is difficult when evaluating the complex tumor microenvironment composed of different cell types (e.g., immune cells and firboblast) [132] as well as the ADME process from multiple organs (gut, liver, kidney, etc.) via drug metabolizing enzymes and efflux transporters [133]. It is important to consider advanced ex vivo models that harbor various cell types, particularly for botanical drugs with mixture components that may target multiple pathways/cells.

Human liver microsomes, containing a wide variety of drug-metabolizing enzymes such as CYPs, are the most budget-friendly ex vivo model to evaluate drug metabolism and food–drug interactions [138]. For example, in the herb-based bioactive flavones study described above, a CYP kinetic analysis was performed on human liver microsomes incubated with various concentrations of CYP substrates in the presence of natural products. The study found that seven CYP isoforms from human liver microsomes are inhibited by the flavones, suggesting that the flavones may interact with drug metabolism in some patients [139]. The cutting-edge technology of human organs on a chip (OoCs), such as human liver on a chip and gut on a chip, are emerging ex vivo models that can bridge the gap between animal studies and clinical trials in predetermining the potential food and drug interactions [140]. However, due to the high costs and complexity of these techniques, there are few published studies of these methods applied to botanical drugs.

Patient-derived organoids (PDOs) are three-dimensional (3D) cultured multicellular clusters derived from the patient’s original tumors. The studies on PDOs and anticancer treatments have been promising and suggest that PDOs may be an ideal platform to identify and assess the efficacy of anticancer medications, ultimately leading to the optimization of treatment plans [141,142]. PDOs have been widely proposed for small-molecule drug development [143], but the application of PDOs in botanical drugs is still in its early stages. Under the framework of immune-oncology, some recent studies have explored the potential of PDOs in tumor immunity by co-culturing PDOs with tumor-reactive T cells or NK cells, suggesting that PDOs can be used to determine if T/NK cells are effective at killing tumors [143,144,145]. There is a major group of botanical drugs classified as immune modulators because they exert immune priming, modulating, and reprogramming functions toward innate and adaptive immunity [146]. In the same vein, PDO and immune cells co-culture models would also be suitable to investigate botanical immune-drugs [143,144,145,146].

#### 3.2.3. In Vivo Models: The Bridge between Cells and Human

Animal models can be manipulated to mimic human disease conditions. Selecting the appropriate animal model that can accurately represent the pathobiology of cancer in the identified human population, as well as choosing the clinically relevant surrogate markers and endpoints with rational experimental design are essential prerequisites to ensure that sufficient and valid evidence is collected from animal studies to move toward clinical development [147]. Conventional models, such as subcutaneous cell-derived xenograft models in immunodeficient mice, are crucial during the early phase of drug discovery to ensure that the molecules have the appropriate pharmacology and activity in the in vivo system [148]. The effect of the drug on the primary tumor can be determined using xenografts derived from patients’ solid tumors (PDX). PDX in immunodeficient mice is widely used to model tumor heterogeneity in patients [149].

However, in the context of immunotherapeutic drug development, xenograft models in immunodeficient mice are not suitable because they lack an intact immune system [150]. Many anticancer foods often exert immune regulatory and/or anti-inflammatory activity, reflecting that innate and adaptive immunity in animal models are essential to validate anticancer immunity induced by botanical-derived drugs [146]. Thus, syngeneic mouse models, which consist of tumor tissues derived from the same mouse strain and retain intact immune systems, are particularly relevant for immunotherapy studies. Ideally, humanized mice and PDX models that harbor patient tumors as well as an intact human immune system can be used to evaluate botanical immuno-drugs [151]. However, the utility of such a model in botanical drug development is limited due to costs and accessibility.

To investigate botanical food intervention in animal models, the administration route and manner in animal models need to be properly designed to mimic drug liberation in humans. There is still a lack of standardization in diet design in food intervention animal studies of cancer [152]. Most food intervention studies administer the foods via the diet, either by incorporation of experimental food ingredients into an open formula semi-purified diet or by addition of a food extract to a standard chow diet [153]. However, many closed-formula natural ingredient chow diets contain unknown compounds that may impact the study endpoints and results. For example, soy is commonly used as a protein source in laboratory chows, and this may inadvertently introduce phytoestrogens into the diet in the form of isoflavones [154]. Open-formula semi-purified diets provide an advantage over standard chow if the food intervention involves a modification of lipids (e.g., corn oil, nuts oil) [155]. For foods with defined active constituents, custom manufacturing the active ingredients into semi-purified diets or into standard chow diets is possible.

Some studies have adopted the oral gavage method, where tubes are used to deliver an exact amount of the food extract to the animal to ensure that the food products are ingested regardless of regular background diet intake [153]. Considering human food intake patterns and human physiological/pharmacological processes for botanical food products (described in Section 2, Figure 2), we cautiously propose oral gavage as the ideal method in animal studies. A counterargument for oral gavage is that it stresses the animals and is limited to water extraction of the foods, but variations in quantity and frequency of food intake among the animals must be addressed and minimized. As such, gavage with a fixed amount/time of testing product ensures a steady quantitative intake from time to time and animal to animal. For the experimental foods with unknown active ingredients, administration of food extract via oral gavage should be considered. Furthermore, in humans, botanical foods may only be delivered through diet in large amounts. Generally, oral administration is more appropriate when botanicals are consumed as pills, capsules, other than bolus doses by humans.

The other variable to account for is the choice of experimental controls. For the treatment-naïve (non-treatment) control, either a control diet without adding experimental food ingredients or vehicles without testing products can be applied to animals in non-treatment control. It is also important to introduce a reference (positive) control, which involves giving a group of animals an approved drug that reflects the presumed action of the testing botanical drug. Furthermore, pharmacology and toxicology indices need to be determined before human studies. This is especially important for food-based interventions, where the dose of whole food extract or food-based natural products in experimental animal studies is generally higher than the dose in a normal human diet [156]. Therefore, justifying the safety of megadose foods in an animal model will be critical. Safety concerns are the primary focus of the assessment, where the following events should be closely monitored: body weight measurements, behavior surveillance (food intake and activity level), target tissue weight, liver/kidney morphology at necropsy, and clinical evaluation of hepatic function (e.g., ALT and AST) [157]. Animal-based pharmacodynamics and pharmacokinetics (PD/PK) are other important aspects of preclinical studies. In PD studies, PD biomarkers that reflect the drug’s effect on the target in an organism are quantitatively measured. PSA is an example of a well-established PD biomarker for drugs targeting the AR receptor signaling pathway in prostate cancers [158]. However, the quantitative PK measurement obtained from systemic exposure is challenging for food-based products because botanical drugs often contain multiple chemical molecules, and the active constituents may not be defined. According to the USFDA guidance, a PD or clinical endpoint assessment may be employed if there are no quantifiable active constituents available for in vivo PK studies [30].

#### 3.2.4. In Silico Models: Multi-Omics and Systems Biology Approaches Reinforce Anticancer Foods Research

As noted above, in anticancer drug treatment, a “one-drug/one-target/one-cancer” approach is problematic because cancer biology is complex on both spatial and temporal dimensions [11]. Additionally, drug responsiveness is affected by network architecture and systemic regulation compared to single molecular targets or cell types [15]. With the emergence of systems biology and multi-omics techniques, the cancer research paradigm has transformed, especially in the field of anticancer drug development [14,15]. Systems biology techniques have influenced wet lab-based preclinical studies and clinical trials in the development of botanical drugs with multiple components, multiple targets, and potential systemic effects [19,20].

A network pharmacology approach supports the development of multitarget anticancer drug development by using advanced computational methods based on systems biology. This approach includes the identification of multiple targets, analysis of functional pathways, computing networks and connectivity, and predicting of drug indications and interactions [159]. Network pharmacology thus opens new avenues for a botanical drug when it comes to understanding the biological systems and the development of the complex bioactive components found in medicinal plants [160]. As we have conducted our research, however, we also have encountered and realized the limitations of such methods. The input data of network pharmacological analysis are the chemical library of the active components in botanical products and the biological library of the targets acting on diseases of interest. For botanical products with defined chemical profiles, network pharmacological approaches facilitate the identification of compound- and disease-related genes, as well as the construction of protein–protein interaction (PPI) network. From there, by using network analysis, key nodes are identified while key biological pathways are predicted. Consequently, additional network validation is performed to successfully validate the interaction between highly active constituents and their putative targets [159,160,161]. However, network pharmacological approaches for botanical foods cannot be achieved given that the chemical constituents are not fully characterized. Therefore, efforts need to be made to thoroughly characterize such chemical profiles via analytical chemistry study and to conduct state-of-the-art single-cell and spatial multidimensional analysis, which would help to define the “net-work-targets, multiple-component-therapeutics” concept.

On the other hand, the interface between “drug” and “food” becomes particularly meaningful when food-derived nutrients are used as therapeutic agents. The concepts and methods of pharmacogenomics and nutrigenomics have been proposed to address individualized medicine and nutrition [161,162]. Both technologies are related to characterizing a wide range of gene variants and single-nucleotide polymorphisms (SNP), which are correlated with a person’s health status along with validating and incorporating genotype-based strategies to optimize health and prevent diseases [161,162]. Pharmacogenomics has been introduced into cancer care and involves characterizing polymorphisms of drug-metabolizing genes (CYPs) and efflux/transport genes (ATP-binding cassettes) that define drug responses. The application of pharmacogenetics narrows the therapeutic index of chemotherapeutic drugs and reduces the risks for life-threatening adverse effects [163]. In nutrigenomics, it is becoming possible to integrate nutrition into therapeutic strategies and interventions for cancer by identifying the “nutrients-gene-cancer” network that underlies cancer initiation and progression [164]. It is crucial that the interface between “food–drug interactions via gene networks” be considered during the development of botanical drugs from botanical foods [165]. Food–drug interaction via the CYPs system is an example of this approach. As explained in Section 2, foods composed of complex chemical mixtures can inhibit and/or induce the activity of drug-metabolizing enzymes and transporters, understanding and assessing these interactions via “omics techniques” and “computer-based approaches” can provide great insights that conventional wet-lab approaches cannot achieve [166,167].

## 4. Critical Considerations in Clinical Studies under IND Regulations

As noted above, there is increasing interest in investigating botanical mixtures as anticancer products, especially from botanical foods. These products are generally thought to have therapeutic activity without significant toxicity [23,27]. However, demonstrating therapeutic efficacy, characterizing the pharmacology, and ensuring consistent quality in clinical trials are particularly challenging. In 2016, the USFDA published a guide, *Botanical Drug Development Guidance for Industry* to advise and facilitate the development of a putative botanical drug. The guide provides critical considerations for clinical trials and recommendations on preparing INDs and new drug applications (NDA) [30]. For botanical products, the typical clinical trial design elements still apply, including study design (single arm or multiple arms), primary objectives and endpoints, target-patient selection (eligibility criteria), IND dosage and administration (treatment plan), scheduled clinical observations and tests, anticipated toxicities, and adverse events. This guide to botanical drugs particularly emphasizes the “totality-of-the-evidence” approach for reliable consistency, and the description of “prior human experience” [28,30,31]. We draw on our experiences in studying white button mushrooms (WBM, Agaricus bisporus) as an anti-prostate cancer IND under USFDA regulation in this section.

### 4.1. The “Totality-of-the-Evidence” Approach for Reliable Consistency of IND

The “totality-of-the-evidence” approach is a concept used to describe how proposed biosimilars should be developed based on a reference medicine. In a stepwise process, analytical, non-clinical, and clinical studies are conducted to determine the quality, safety, and efficacy of a proposed biosimilar. Similarly, to ensure therapeutic consistency, the “totality-of-the-evidence” approach for a food-based botanical drug requires a manufacturing process that includes rigorous quality control (QC) and the development of clinically relevant bioassays [28,30,31]. As discussed in Section 3.1, a clinically relevant bioassay that reflects the product’s intended mechanism of action and/or relates to a surrogate biomarker of a certain cancer is desired due to the heterogeneous mixture of botanical INDs which often lack quantitative chemical QCs [28,30,31].

### 4.2. Choosing the Appropriate Patient Population

Ideally, clinical study participants for botanical drug-based cancer treatment should be from the intended target population. It is important to justify drug indications and target populations based on “prior human experience” as well as the mechanism-based, clinically relevant preclinical studies [28,30,31]. It is critical to understand the chemical and biological potency of a botanical product when applying it to cancer patients with various histological and molecular subtypes. Unlike chemically synthetic molecules, natural molecules from botanical medications may act as weak partial agonists/antagonists of therapeutic targets, which would lead to fewer side effects. In terms of safety versus efficacy pertaining to patient selection, patients with low-risk or low-grade cancer (e.g., low-risk prostate cancer, low-risk differentiated thyroid cancer, or low risk of breast ductal carcinoma in situ) that have a low chance of metastasizing may benefit from using botanical products as their initial cancer treatment [168]. For cancers with aggressive histological or molecular subtypes, receiving harsher treatment as early as possible would be needed for a better prognosis. As a result, evaluating cancer risk based on histological features and molecular markers, coupled with the understanding of the chemical and biological properties of food products, help to establish the feasibility of applying botanical foods to cancer patients.

### 4.3. Critical Safety Concerns in Early-Phase Clinical Studies

“Safety versus efficacy” is a major source of failure in clinical studies, especially in early-phase clinical trials where safety evaluation is the priority [169]. It is pertinent to note that often the initial evidence cited for undertaking clinical trials of food-based products for cancer treatment comes from low/regular dietary food intake in epidemiologic nutritional investigations. However, anticancer interventional studies regularly introduce larger doses of food-based products (whole extract or supplements) that are taken in the diet. Knowledge of pharmacodynamics and pharmacokinetics (PD/PK) is essential in evaluating the safety of any anticancer medication, including those from natural sources. This is often done in the context of Phase I clinical trials. The quantitative PK measurement of systemic exposure is challenging for food-based products because botanical drugs such as WBM contain multiple chemicals, and the therapeutically active constituent may not be known [30]. In most cases, PD markers are also surrogate endpoint markers for determining the link between drug dose, target effects, and biological tumor response since they reflect the drug’s effect on the target in an organism [170].

In addition, food–drug interactions can be antagonistic, where a natural product could inhibit certain drugs’ metabolism, leading to dose-related toxicities. In some cases, a plant-derived product may contain molecules that act as competitive antagonists or agonists for certain receptors [171]. An example of a weak partial agonist is genistein, a soybean flavonoid that blocks estrogen receptors. In the presence of tamoxifen, genistein can act as an agonist to activate the receptor, supporting the use of small doses of this food-derived product as a medication for breast cancer [172]. However, there is insufficient evidence to support the use of phytoestrogens in the entire general population, including patients with benign breast disorders, those at risk of breast cancer, and even cancer survivors [172].

Lastly, the objectives and endpoints should be carefully considered when formulating a clinical trial. Clinical trials typically have a primary objective or endpoint, where other objectives and endpoints are added to address the safety and efficacy of the drug. However, different phases of trials have different objectives. An early-phase clinical trial assesses a drug’s safety and tolerability, whereas a late-phase trial assesses its efficacy [173]. Endpoints must be established to measure objectives in each stage of clinical development. Ideally, the endpoints should be clinically relevant, sensitive to intervention effects, practical to measure, and unbiased [174]. For botanical drugs that have heterogeneous chemical mixtures and many potentials for systemic effects, multiple endpoints should be considered to address the objectives of the trials.

### 4.4. Diversity of Research Participants and Their Dietary Background

Recently, the USFDA has issued new draft guidance on a diversity plan to enroll participants from different racial and ethnic populations when proposing clinical trial designs for investigational products. The USFDA states that “the diversity plan” will be considered “an important part” of the product’s development program. Enhancing clinical trial diversity is ethical and scientifically imperative [175]. Due to the ethnic heterogeneity of the US population, patients with different genetic backgrounds may have different safety and efficacy signals, even when they receive the same therapy. The lack of diversity may yield a biased understanding of the safety and efficacy of novel therapies across population subgroups.

In food-based clinical studies, diversity is particularly meaningful and essential, given the assortment of dietary preferences, general health conditions, and gut microbiomes among people from different cultures and races [176]. A botanical food’s or a botanical drug’s PK properties may be affected by the gut microbiome and diet by interfering with absorption, metabolism, and activity, ultimately affecting how effective the product is against cancer [177].

### 4.5. The Challenges of Using a Placebo in Botanical Foods Intervention Studies

Designing and using placebo controls are critical in clinical studies. However, in food intervention studies, it is occasionally often impractical or impossible to design a placebo product that is physically identical to the investigational drug without sacrificing sensory qualities. Moreover, for the botanical food products of which the active constituents are not known, it is technically unachievable to create placebo products without pharmaceutical activity. We have reviewed several published reports that address which products were suitable for placebo controls in food intervention studies [178,179]. However, based on available literature botanical products that are suitable for use as a placebo in food interventions are largely lacking [178]. The specific guidelines or standards of botanical food-placebo design, including usage requirements, preparation specifications, quality assessments, and reporting guidelines should be developed further to improve their usage [179]. The USFDA’s guidance suggests consulting their regulatory office for using of such placebo in botanical foods intervention studies [30].

## 5. Case Study: Development of White Button Mushroom as Anticancer Botanical Medicine at the City of Hope

At the City of Hope, our laboratory has more than 20 years of experience and has established records in defining anticancer foods under both preclinical and clinical settings. The foods under investigation include but are not limited to WBM [180,181,182,183,184,185], grape seed extract [186,187,188], pomegranate [189], blueberries [190,191], *Eugenia jambolana Lam* berry [192]. WBM products (WBM tablets generated from fresh freeze-dried WBM) were approved as IND by the USFDA for clinical intervention studies. We have completed phase I trials in prostate cancer (NCT00779168) and breast cancer (NCT00709020). A phase II trial for prostate cancer (NCT04519879) and a prevention trial for women with a high risk of breast cancer (NCT04913064) are currently open for recruitment.

### 5.1. “Prior Human Experience” of Mushroom Products as An Anticancer Medicine

In many cultures, edible or medicinal mushrooms are used as traditional herbal medicine. Several mushroom species have been shown to exhibit anticancer effects in vitro, in vivo, and also in humans [193]. There is extensive “prior human experience” with using the mushroom-derived product as anticancer medicine, suggesting that bioactive ingredients in mushrooms may play a significant role in cancer prevention, reducing the risk of recurrence, and therapy [194,195]. A large cohort, long-term follow-up study in 36,000 Japanese men over decades suggests an association between eating mushrooms and a lower risk of prostate cancer [194]. Another meta-analysis from more than 19,500 cancer patients also suggested a strong association between higher mushroom consumption and a lower risk of cancer, particularly breast cancer [195]. WBM is the most common and budget-friendly edible mushroom in North America. Our laboratory has thus pioneered preclinical and clinical studies for WBM on breast cancer [[180],[182] and prostate cancer [[181],[183],[184],[185].

### 5.2. Clinically Relevant In Vitro Bioassay Established in Our Lab

Our research journey has led us to establish a series of robust clinically relevant bioassays targeting sex-hormone receptors and steroidogenesis enzymes to evaluate whole-food extracts against breast and prostate cancers. The first line of estrogen receptor (ER)-positive breast cancer treatment in the clinic is endocrine therapy, where aromatase inhibitors (AIs) and antiestrogen drugs, respectively, block estrogen formation and ER activity [196]. To mimic these molecular effects in the laboratory, we have established an ER-positive breast cancer cell line overexpressing aromatase, MCF-7aro, to screen for possible aromatase inhibitors [197]. From such bioassays, we have determined that extracts of the aforementioned foods [180,186,189,192] as well as phytochemicals [135,136,137,138] inhibit aromatase activity, thus suppressing estrogen-dependent breast cancer cells. A next-generation MCF-7aro/ERE cell model-based luciferase assay has been successively developed to simultaneously screen aromatase inhibitors, ERα ligands, and estrogen-related receptor alpha (ERRα) ligands,. MCF-7aro/ERE is derived from MCF-7aro by stably transfecting with pGL3(ERE)3-Luc, a promoter-reporter plasmid that contains ERE (estrogen-responsive element) and a luciferase expression reporter. Additionally, ERE-mediated luciferase gene expression reporter activity is quantified through a luminescence-based approach [198]. Known as “AroER Tri-Screen”, this robust and biologically relevant assay is used to characterize AIs, ER agonists, and/or ER antagonists found in the environment (environmental endocrine disruptors) [199,200] or in natural products [200].

Another bioassay has been developed to screen for molecules interacting with LNCaP cells, which is an androgen receptor (AR)-positive and prostate-specific antigen (PSA)-expressing prostate cancer cell line. We have used an LNCaP-PSA luciferase assay to screen for androgen receptor (AR) inhibitors from both USFDA-approved compounds and natural products [201]. The activation of AR is the driving force of prostate cancer. In contrast, PSA serves as a diagnostic and surrogate endpoint marker for pharmacological strategies against AR activation in prostate cancer [158]. The LNCaP-PSA luciferase assay has been developed by transfecting a PSA promoter-firefly luciferase plasmid into LNCaP, an AR-positive prostate cancer cell line. As a result of the transfection, AR activation-induced PSA promoter activity is measured by luminescence signal [201].

By establishing clinically relevant in vitro bioassays for breast cancer (AroER Tri-Screen assay) and prostate cancer (LNCaP-PSA luciferase assay), we have shown that WBM extracts significantly inhibit aromatase activity and ER activation in breast cancer cells [180], and also interrupt the AR activation and PSA expression in prostate cancer cells [184]. We have also identified a series of ER antagonists, Ais, and AR antagonists via the above assays using over 400 purified natural product compounds from the National Cancer Institute/NCI Natural Products Repository.

### 5.3. Various Preclinical Models and MultiTarget Profiling Approaches Applied in Our Study

Our lab applied various preclinical models and multitarget profiling approaches to define the biological activities that WBM exerts. The preclinical models included cell lines, PDX models, syngeneic mouse models et al. State-of-the-art techniques such as nuclear receptor coregulator interaction assays, single-cell and spatial multidimensional profiling analysis, cytokine array, and multiplex flow cytometry were applied.

In our study of the antiandrogenic activity of WBM in prostate cancer, 5 prostate cancer cell lines (LNCaP, VCaP, PC-3, DU-145, and 22RV1) that have captured histological (prostate adenocarcinoma) and molecular features (AR-positive, AR-negative, AR-variants) of human prostate cancers are selected and 1 prostate normal epithelial cell line (RWPE1) have been used as in vitro models [181,184]. We have used both subcutaneous cell-derived xenograft models (DU-145 and PC3 prostate cancer cells engrafted in NSG mice) [181] and AR-positive/PSA-expressing prostate cancer PDX model (TM00298 PDX tumors engrafted in NSG mice) as in vivo models [184]. The tumor growth rate and tumor volume are evaluated as the responsive index. We also have measured PSA expression as a pharmacodynamic (PD) biomarker in PDX models [184]. In such studies, we also have introduced enzalutamide, an USFDA-approved synthetic AR antagonist for prostate cancer, as the reference drug. From there, we have identified that WBM extract exerts antiandrogenic activity, which is phenotypically similar to the therapeutic effect of enzalutamide [184]. Conjugated (9Z, 11E)-linoleic acid (CLA-9Z11E) is an active component that we have identified in the WBM extract [180,181]. By conducting AR Coactivator Interaction Assays for a direct interaction of CLA-9Z11E with AR, we have found that CLA-9Z11E exerts a strong antagonistic potency against the AR coactivator [184]. The knowledge gained from this study indicates that WBM intake affects prostate cancer by interfering with the AR signaling axis.

In another study about the immunomodulatory effects of WBM in prostate cancer, the syngeneic mouse model [mouse prostate cancer cells (TRAMP and Myc-CaP cells) allografted in C57BL/6 or FVB mice, respectively] has been used. We have introduced Lentinan, the bioactive β-glucans and potent immunomodulator isolated from Shiitake mushrooms, as the reference drug [185]. Lentinan has been approved and used as an adjuvant therapeutic drug for multiple types of cancer in Japan and China [202,203]. By feeding immunocompetent mice with WBM extract and Lentinan, we have demonstrated that the cytokine signatures associated with treatments of WBM and Lentinan greatly overlap. We also have observed that both WBM and Lentinan have decreased the total counts of MDSCs in peripheral blood and spleen. Our results have implied that β-glucans are probably a major component of WBM that modulates immune function [185]. As mentioned previously, we are further exploring WBM-induced immune-mediated anticancer effects with a state-of-the-art single-cell and spatial multidimensional profiling analysis.

### 5.4. The “Totality-of-the-Evidence” Approach for WBM Product as IND

The “totality-of-the-evidence” approach is a rigorous quality control (QC) process that is recommended by the USFDA for food-based botanical drugs [28,30,31]. We use this approach in phase I/II trials treating prostate cancer with WBM. Freshly grown WBMs are collected and processed into a freeze-dried powder at a US-based farm that is Good Agricultural and Collection Practices (GACP) and Good Manufacturing Practices (GMP) compliant. The mushroom farm’s in-house laboratory carries out initial QC on raw material (freeze-dried WBM powder in our study), including physical properties (particle, moisture), chemical hazards (heavy metals), and microbiological contamination (yeast, mold, E. coli, salmonella, etc.). The raw materials are then shipped to a third-party facility to produce WBM tablets in accordance with GMP standards. Finally, we perform the LNCaP-PSA luciferase assay and qRT-PCR of PSA expression as the mechanism-based and clinically relevant bioassay quarterly to document the stability and bioactivity of WBM tablets (IND).

### 5.5. Designed and Conducted Clinical Interventional Studies on Human

Based on “prior human experience” and mechanism-based clinically relevant preclinical studies, our team at the City of Hope has conducted the first single-arm open-label clinical phase I trial of WBM tablets in 36 patients with biochemically recurrent (BCR) prostate cancer. It is a trial without a placebo control, in which both the investigators and the patients are aware of the drug being given while the patients in the control groups do not receive WBM tablets [183]. In this trial, we have evaluated the feasibility, toxicity, and biological activity of prolonged therapy with WBM tablets. Phase I is designed as a dose-escalation study, with six different dose levels of the mushroom product given to patients with BCR prostate cancer after previous local therapy. PSA is used as a PD marker to determine the WBM drug effect [183]. Two patients have exhibited a complete response, with a reduction in PSA that reaches undetectable levels, and two other patients have experienced partial clinical and PSA responses, while the other thirteen patients have some reduction in PSA from the baseline. Responsive patients have also experienced a decline of myeloid-derived suppressor cells (MDSCs) in blood circulation, which is correlated to PSA reduction [183]. Taken altogether, these findings suggest that bioactive chemicals in WBM may exert both antiandrogenic and immunomodulatory effects.

Based on our phase 1 trial findings, we have conducted several reverse-translational studies to understand the various biological activities that WBM exerts. Through those studies, we have demonstrated that WBM contains AR antagonists that can block the activation of AR signaling, which is the driving force of prostate cancer progression [184]. The knowledge gained from this study indicates that WBM intake affects prostate cancer by interfering with the AR signaling axis. Additionally, we have learned from our preclinical studies and previous human-based evidence that WBM may also contain immune-regulatory and anti-inflammatory chemicals that may synergistically contribute to their anti-prostate cancer properties [185].

We thus have designed two cohorts in our currently ongoing open-label randomized control (RCT) phase 2 trials (NCT04519879) using WBM. Patients with BCR prostate cancer (cohort 1) and low-risk prostate cancer (cohort 2) are chosen as the targeted populations. The neuroendocrine type of prostate cancer is more aggressive and is excluded from our clinical trial [183]. Particularly for those patients with low-risk cancer, the growing clinical evidence indicates that administering more aggressive treatment such as surgery or radiation at the early cancer stage does not necessarily prevent men from having a poor outcome and can result in serious urinary and/or sexual side effects [204]. As a result, many patients with low-grade and low-risk prostate cancer are advised to be on active surveillance to avoid the side effects associated with these aggressive treatments [205]. Another scenario is BCR prostate cancer patients who experience an increase in PSA after treatment with surgery or radiation. Regarding drug therapy, androgen deprivation therapy (ADT) remains the regimen of choice, however, the side effects of ADT have a significant negative impact on the quality of life [206]. We have hypothesized that for patients with low-risk prostate cancer under active surveillance or BCR prostate cancer, the use of botanical products that contain natural-form bioactive molecules will be a safe alternative treatment that can prevent or delay the progression to advanced disease [207]. In considering that WBM has both antiandrogen and immunoregulatory activity, multiple endpoints are designed to explore various functions of WBM. PSA is the surrogate endpoint marker for pharmacological strategies against AR activation; PSA progression time is a secondary endpoint to predict disease progression. Furthermore, an exploratory objective for our phase II trial is to measure changes in immune mediators such as cytokines or immune cells. Moreover, our ongoing phase 2 trial are aiming to enroll participants from different racial and ethnic groups. The efforts include delivering the flyers in different languages, reaching different cultural communities, etc.

The information gained from the aforementioned studies has greatly improved our overall understanding of how WBM may contribute to the prevention and treatment of prostate cancer while serving as an important scientific basis for the future development of WBM as anticancer medicine.

## 6. Discussion and Outlook

There is an enthusiastic public interest in the idea of anticancer superfoods since these foods are considered by many in the public to be safer than synthetic drugs because anticancer superfoods are natural products. Following a diagnosis of cancer, nearly half of American patients begin taking food-derived products or seek advice on consuming these products [208]. However, health professionals are reluctant to offer actionable advice to patients on such products because they doubt their efficacy and safety. Generally, these concerns include but are not limited to (1) whether food products should be incorporated into a treatment plan; (2) which products are safe and effective to use as treatments; and (3) whether food products are antagonistic to standard treatment with chemotherapy or immunotherapy. The most fundamental reason for the hesitation in clinical decision-making by physicians is the lack of science-based and clinically proven results demonstrating the efficacy of anticancer superfoods. Mechanism-based and clinically targeted development of plant foods as cancer therapeutics are needed by both the public and health professionals.

Among the controversies about anticancer foods is whether they should be consumed as whole foods or as dietary supplements derived from them. It is a concern that the manufacturing and application of dietary supplements in cancer care are largely unregulated [209] According to the scientific literature, plant-derived functional foods may target multiple oncogenic signaling pathways and cellular compartments at the same time via heterogeneous molecules that are present in plants, rather than by an isolated chemical constituent [23,24,25,26,27,30,31,32,33,34,35]. By using the network pharmacology approach, it has been shown that phytochemical mixtures in whole foods may have networking (additive or synergistic) effects on cancer [210,211]. From our studies on WBM, we also demonstrated that WBM provides antiandrogen effects via CLA-9Z11E and immunoregulatory functions through β-glucans. Isolated CLA-9Z11E or β-glucan exhibited either antiandrogen or immunoregulatory effects whereas WBM whole food extract have exhibited both properties simultaneously [184,185]. Another example is our study on grape seed extract (GSE) in breast cancer models. GSE has suppressed tumor angiogenesis in breast cancer xenograft tumors by interfering with the vascular endothelial growth factor (VEGF) signaling pathway. Interestingly, when the bioactive polyphenol in GSE has been depleted, the extract loses its antiangiogenesis activity [187,188]. It is possible for single compounds to have weaker anticancer effects or lose their bioactivity without coexisting with other molecules. Pharmacology/toxicology properties, bioavailability, and dosage are the most critical considerations for isolated compounds [212]. However, one has to admit that it is easier to standardize the production and regulation of purified bioactive components with defined physical, chemical, and biological characteristics [209]. It remains unclear whether whole food extract or the bioactive ingredient should be considered as the anticancer candidate agent [213]. Future studies that assess botanical food products should consider these criteria (Figure 1 and Table 1) to define the whole food product and/or isolated compounds as an anticancer botanical drug.

Another trending discussion with great clinical interests involves combining botanical products as (neo)adjuvant treatments or concurrently with conventional cancer treatments such as chemotherapy, radiotherapy, and immunotherapy [214,215]. Such combinations and possible additive or synthetic interactions with conventional drugs may improve the outcome and reduce risks. However, this means that physicians will need to be fully aware of the benefits and especially the risks that are potentially caused by food–drug interactions [165,166,167]. As we have discussed in Section 2 (Table 2), experimental evidence for such interactions is well defined and documented in the literature but the same cannot be stated for the botanical drugs with unknown active ingredients. A major concern about the latter is the fact that their food/herb–drug interactions are still theoretical [94]. It is noteworthy that newly published preclinical and clinical studies have reported on the benefits of combining botanical drugs and conventional treatments, such as reducing chemotherapy-related side effects [216] and overcoming chemoresistance [217]. On an even more positive note, among the fields that are advancing rapidly is the development of botanical immune drugs in cancer therapy [146]. In our research on the WBM, we have learned that the botanical polysaccharide (i.e., β-glucan) in mushrooms or yeast exerts potent immune-regulatory activity by reprograming immune cells and signaling. Therefore, yeast- or mushroom-derived β-glucan have been actively proposed as (neo)adjuvant treatments to enhance or prime the responsiveness of cancer immune therapy [218,219]. These efforts include both preclinical and clinical studies by combining mushroom or yeast-derived polysaccharides with immune checkpoint inhibitors such as anti-PD1/PD-L1 [220,221].

As a final note, we hope that this article will provide researchers, health care providers, and industry sponsors with landscape-based perspectives and insights. This article is for those who wish to understand how botanical foods and their derivatives contribute to cancer pathobiology, as well as develop relevant therapeutic approaches to treat cancer by conducting clinically oriented and evidence-based translational studies under USFDA regulatory frameworks.

## Figures and Tables

**Figure 1 cancers-15-00701-f001:**
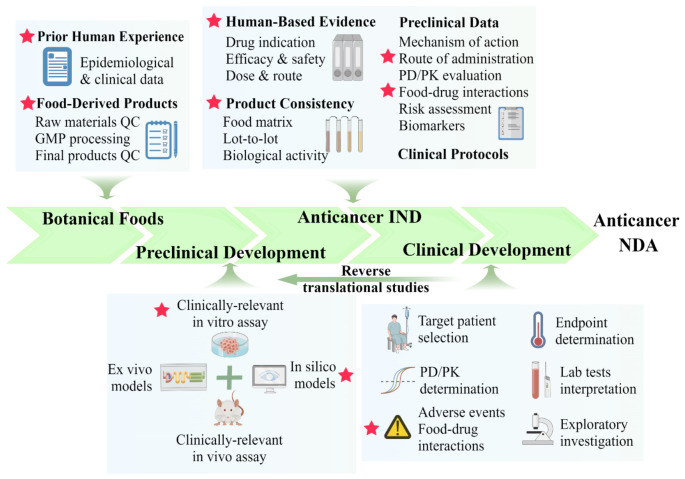
Stepwise development pipeline of botanical foods as anticancer medicines. Abbreviations: QC, quality control; GMP, good manufacturing practices; PD/PK, pharmacodynamics/pharmacokinetics; IND, investigational new drug; NDA, new drug application. Pentagrams label the unique steps that applied to botanical food development compared to conventional drug development (the figure is prepared with Figdraw, https://www.figdraw.com, accessed on 10 January 2023).

**Figure 2 cancers-15-00701-f002:**
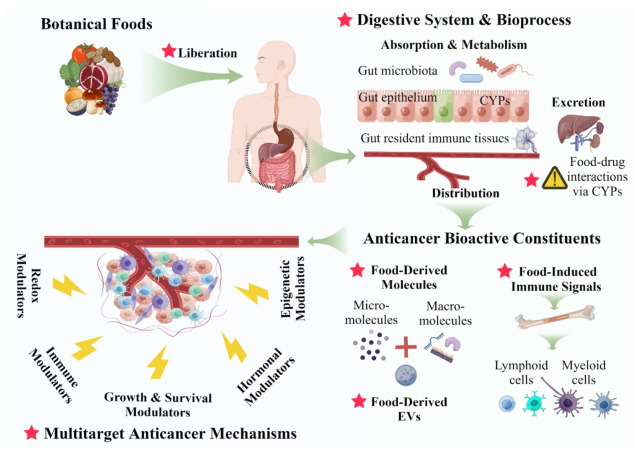
The physiological and pharmacological processing of botanical foods as anticancer medicine in cancer patients. Pentagrams label the unique processes that are applied to botanical food development compared to conventional drug development Abbreviations: CYPs, cytochrome P450 enzymes; EVs, extracellular vehicles. (The figure is prepared with Figdraw. https://www.figdraw.com, accessed on 18 January 2023).

**Table 1 cancers-15-00701-t001:** Seven critical checkpoints in the developing anticancer botanical drugs from plant-based foods.

No.	Checkpoints	Explanations
**1**	**Propose hypothesis based on “prior human experience”**	The documented history of human data would allow the researchers and regulatory agents to proceed directly into a clinical evaluation of efficacy and safety. This could potentially shorten expensive preclinical development efforts, as well as reduce the likelihood of development failure.
**2**	**Establish mechanism-based and clinically relevant in vitro bioassays**	A bioassay that reflects the drug’s presumed mechanisms of action or measures the drug-responsive cancer marker should be developed if the active constituents of foods cannot be quantified chemically. This would help assess product batch quality and activity.
**3**	Choose clinically representative animal models for testing	Selecting a model that reflects the pathobiology of the cancer (carcinogenesis, progression, and metastasis) as well as the clinically relevant biomarkers and endpoints that can provide supportive evidence for future clinical development.
**4**	**Perform rigorous quality control measurements and assays to ensure the therapeutic consistency**	Therapeutic consistency is maintained, including the source and quality of botanical raw materials, assuring manufacturing processes are GMP compliant, and performing extensive chemical and biological characterization.
5	Design clinical trials rationally by selecting the target population with evidence-based indication and clinical need	For both ethical and safety concerns, lower-risk cancer patients are recommended. For the botanical food product with clinical indications and an expected mechanism of action, first-in-human studies should focus on toxicity determination and proof-of-concept target engagement PD studies, while randomized control trials (RCTs) may be preferred in Phase II/III study.
**6**	Conduct reverse translational studies to discover mechanisms underlying patient outcomes and clinical observations.	Reverse translational studies, also known as bedside-to-bench research, allow for seamless and cyclical research, where observing patient responses can stimulate new hypotheses that may help refine and guide bench research that can lead to future clinical trials.
**7**	**Introduce advanced multitarget and computational characterization techniques**	Botanical foods can target multiple pathways simultaneously acting on a large response network. Using a single biochemical assay or bulk analysis of genes/proteins has insurmountable limitations. The development of computer-based network pharmacology along with state-of-the-art single-cell and spatial multi-omics analysis would help to define the “network-targets, multiple-component-therapeutics” concept.

**Table 2 cancers-15-00701-t002:** The clinically advisable or actionable databases on “anticancer foods” for researchers, industry sponsors and health professionals.

Name of Database	Food/Supplement	Information Provided	Reference
National Cancer Institute (NCI) Complementary and Alternative Medicine Cancer Therapy Interactions with Food and Dietary Supplements (Physician Data Query, PDQ^®^)	Over 200	General information on foods/supplements List of antioxidants, herbs, and foods Foods, dietary supplements, and cancer drug interactions Interactions of foods with cancer therapies Food and dietary supplement summary	[41]
Memorial Sloan Kettering Cancer Center Integrative Medicine About Herbs, Botanicals, & Other Products	Over 290	Clinical summary Food source Purposed uses and benefits of food and herbs Mechanism of action, herb–drug, and herb–lab interactions Contraindications and adverse reactions	[42]
The Natural Medicines Comprehensive Database for Foods and Herbs	Over 1200	Overview of foods and herbs Warnings, safety, adverse effects, dosing, administration, and effectiveness of foods and herbs Drug/Supplement interactions Condition/lab interactions Pharmacokinetics and overdose Mechanism of action	[43,44,45]
NCI Drug Dictionary	Over 200	Scientific names of food and herbs General studies and the description of food/substance Potential benefits and adverse effects Active clinical trials using the agent	[46]
US Department of Agriculture Food Data Central Foundation & Experimental Foods	Over 700	The chemical composition of food Analytical methodology Food procedures research purpose Study design, results, and supplemental information	[47]
American Institute for Cancer Research AICR’s Foods that Fight Cancer™	26 types of plant-based foods	Food ingredients Ongoing area of investigation in labs and humans “Convincing” or “probable” evidence “Limited suggestive evidence”	[48]

**Table 3 cancers-15-00701-t003:** Ten “anticancer foods”: interactions with drugs and review of evidence.

Foods	Food–Drug Interaction	Level of Evidence	References
Edible Mushroom	Cyclophosphamide	In vitro or animal study	[39,40,41,42,43,44,45,46,47,48,52,53]
	Tamoxifen	In vitro or animal study	
	Cytochrome P450 2C9 and 3A4	In vitro or animal study	
	Immunosuppressants	Theoretical based on pharmacology	
	Antidiabetic drugs	In vitro or animal study Lower-quality randomized controlled trial Theoretical based on pharmacology	
	Warfarin	Anecdotal evidence	
	Antihypertensive drugs	Theoretical based on pharmacology	
Pomegranate	Cytochrome P450 2C9 and 3A4	Nonrandomized clinical trial	[39,40,41,42,43,44,45,46,47,48,54,55,56,57,58,59,60]
	Cytochrome P450 1B1 and 2D6	In vitro or animal study	
	ACE inhibitors	Nonrandomized clinical trial	
	Antihypertensive drugs	Nonrandomized clinical trial	
	Rosuvastatin (Crestor)	Anecdotal evidence	
	Warfarin (Coumadin)	Anecdotal evidence	
	Carbamazepine (Tegretol)	In vitro or animal study	
	Tolbutamide (Orinase)	In vitro or animal study	
Grape	Cytochrome P450 1A2, 2C19, 2D6, 2E1, and 3A4	Lower-quality randomized controlled trial In vitro or animal study	[39,40,41,42,43,44,45,46,47,48,61,62,63,64,65,66]
	Cytochrome P450 2C9	In vitro or animal study	
	Cyclosporine (Neoral, Sandimmume)	Lower-quality randomized controlled trial	
	Phenacetin	Lower-quality randomized controlled trial	
	Anticoagulant/Antiplatelet drugs	In vitro or animal study	
	Midazolam (versed)	In vitro or animal study	
Cinnamon	Cytochrome P450 2C9, 3A4, 2A6, and 2D	In vitro or animal study	[39,40,41,42,43,44,45,46,47,48,67,68,69,70,71]
	Antidiabetic drugs	Lower quality randomized controlled trial	
	Hepatotoxic drugs	Theoretical based on pharmacology	
	Statins	Case–control study	
	Pioglitazone	In vitro or animal study	
Garlic	Cytochrome P450 2E1 and 3A4	Nonrandomized clinical trial, lower quality randomized controlled trial	[39,40,41,42,43,44,45,46,47,48,72,73,74,75,76,77,78,79]
	Cytochrome P450 2C9 and 2C19	In vitro or animal study	
	Tacrolimus (Prograf)	Case–control study	
	Antidiabetic drugs	Lower-quality randomized controlled trial	
	Protease Inhibitors (Darunavir, Saquinavir)	Nonrandomized clinical trial	
	Anticoagulant/Antiplatelet drugs	Theoretical based on pharmacology	
	Antihypertensive drugs	Theoretical based on pharmacology	
	Atazanavir (Reyataz)	Anecdotal evidence	
	Isoniazid	In vitro or animal study	
	Warfarin (Coumadin)	Anecdotal evidence	
	Insulin	In vitro or animal study	
	P-Glycoprotein substrates	Clinical cohort study	
Broccoli	Cytochrome P450 1A2 and 2A6	Lower-quality randomized controlled trial	[39,40,41,42,43,44,45,46,47,48,80,81]
Tomato	Cytochrome P450 1A2, 2C9, 2D6, 2E1, and 3A4	Clinical cohort study In vitro or animal study	[39,40,41,42,82]
Walnut	Cytochrome P450 2A2, 2B1, 2B2, 2C6, 2C11, and 3A1	In vitro or animal study	[39,40,41,42,83]
Ginger	Cytochrome P450 2C9, 2C19, 2D6 and 3A4	In vitro or animal study	[39,40,41,42,43,84,85,86,87,88,89,90,91]
	Anticoagulant/Antiplatelet drugs	Nonrandomized clinical trial	
	Nifedipine (Procardia)	Nonrandomized clinical trial	
	Warfarin (Coumadin)	Lower-quality randomized controlled trial	
	Losartan (Cozaar)	In vitro or animal study	
	Phenprocoumon (Marcoumar)	Anecdotal evidence	
	Antidiabetic drugs	In vitro or animal study	
	Calcium channel blockers	In vitro or animal study	
	Cyclosporine (Neoral, Sandimmune)	In vitro or animal study	
	Metronidazole (Flagyl)	In vitro or animal study	
	NSAIDS (diclofenac or ibuprofen)	Clinical cohort study	
	Tacrolimus	In vitro or animal study	
	Cyclosporine	In vitro or animal study Clinical cohort study	
Berries	Cytochrome P450 1A1, 2A2, 3A1, 2B1, 2B2, 2C6 and 2C11	In vitro or animal study	[39,40,41,42,43,92,93,94,95,96]
	Buspirone (BuSpar)	Nonrandomized clinical trial	
	Flurbiprofen (NSAID))	Nonrandomized clinical trial	
	Antidiabetic drugs	In vitro or animal study	

## Data Availability

Not applicable.

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
