# Peer review of "Mechanism-Driven and Clinically Focused Development of Botanical Foods as Multitarget Anticancer Medicine: Collective Perspectives and Insights from Preclinical Studies, IND Applications and Early-Phase Clinical Trials"

_cancers, 2023, doi:10.3390/cancers15030701_

Round 1
Reviewer 1 Report
This paper discusses a range of considerations in the development of botanical foods for therapeutic use in cancer patients. The focus appears to be on how to do this in a manner that is complaint with FDA regulations and results in an IND that can be used in clinical trials. The overall objective of the paper is important and adds to the existing literature. Of particular importance are the comments about interactions of botanical foods and their constituents with anti-cancer and other drugs and drug metabolizing enzymes. There are, however, a number of concerns with the paper that detract from its merit.
1. The paper includes quite a number of discussion points that are generally applicable to development of cancer drugs/treatments. In particular, the description of issues related to in vitro, in silico, and in vivo models is very general and does not critically identify how the application of these models for the study of botanicals differs from investigations of single compounds/drugs. For example, it is not explained how to test botanicals, which by nature are mixtures, in in vitro models in a relevant manner and efficacy testing is not really addressed at all. And for the in vivo models section, there is no discussion of how to treat the animals with the botanicals in a relevant manner; how to incorporate botanicals into the diet and the role of the diets used, for example, are not addressed. The use of silico models may be particularly applicable to the study of anti-cancer activity of botanicals, but again there is no discussion about how to test botanical mixtures in such models.
2. In both the in vivo models and clinical trials sections there is no discussion of the range of considerations of the all-important use of placebo controls. Even the word placebo is not found in the paper. Also not well addressed are considerations related to how to combine botanicals with conventional cancer treatments; this could be done as (neo)adjuvant treatments or concurrently with conventional treatments in which case possible interactions are extremely import issues.
3. The authors use their experience with the development of botanicals as cancer treatments as examples, with a strong focus on their studies of white button mushrooms and of prostate cancer. However, the description of these studies is intermingled with various other discussions which makes it difficult to read some parts of the paper. The authors may want to consider creating a separate section that illustrates the general points they raise with their experience with these mushrooms and prostate cancer studies.
4. The botanical foods listed in Table 2 and Supplemental Table 1 were reportedly selected because they are “the ten most-reported botanical foods that have clinically proven anti-cancer activity”. However, none of these ten foods have been shown to have clinically proven anti-cancer activity as far as this reviewer can determine; no references are provided for such clinically proven anti-cancer activities. All have biological activities that may or may not relate to anti-cancer activity in model systems (mostly in vitro models), but no models are listed in Supplemental Table 1 that have cancer as endpoint such as reduction of growth of xenografted tumors or PDX grafts. These are major concerns. The authors may want rethink the selection of foods and include anti-cancer efficacy in their tables.
5. There are also some concerns with the Tables and Figures: In Table 1, the references should be included; this is only done in the text of the paper. Furthermore there are a number of confusions and not well phrased sentences in this table with the following as example for the NCI database:
Foods, Dietary Supplements, and Cancer Drug Interaction – this is duplicative with the next line:
Summary of the Evidence for Cancer Therapy Interactions with Foods and Dietary Supplements – this should be turned around: “interactions of foods with cancer therapies”
Changes to This Summary – what summary is meant here?
About This PDQ Summary – what does this mean?
This is only one example of many such inconsistencies.
In Table 2, there sometimes is and many times is no horizontal connection between the second and third column; this is very confusing. Furthermore, the same five references are indicated for every item in the list which greatly reduces the usefulness of the references and the information in the table for the reader; the authors may want to reconsider how this is done.
In Figure 1 the abbreviation EVs should be explained in the legend. Also, in this figure it would be important to mark those steps that are unique or different for botanical food from those that pertain conventional anti-cancer drugs. This could also be done in Figure 2 and in Table 3 where standard steps in anti-cancer drug development as well as botanical-food specific steps are intermingled.
6. There are several typographical errors that require careful proofreading and correction. In addition, there are several errors in style and grammar that require careful editing of the text.
Author Response
We acknowledge the reviewer’s constructive comments. The reviewer provided detailed suggestions in areas where we could improve this manuscript. The suggestions have helped to improve the accuracy, integrity, and readability of our manuscript. We have carefully considered and incorporated the recommendations into the revised version accordingly.
Please see the attachment.

Reviewer 2 Report
In this manuscript, the authors comprehensively reviewed how to conduct preclinical and clinical studies of the anti-cancer activity of botanical foods. The authors provided a novel perspective: They didn’t focus on summarizing the scientific conclusions of the functions of botanical food in cancer in previous literature. Instead, they criticized the limitations in the methodology of designing and conducting preclinical/clinical studies of botanical food anti-cancer research and gave suggestions based on their experiences in this field. As mentioned in the manuscript, most botanical food anti-cancer studies performed limited research on cancer cell lines and lacked evidence from animal models or human clinical trials. Since these studies aim to promote the clinical application of botanical food as a treatment for cancers, a reasonable research methodology following the guidance of USFDA is necessary to conduct high-quality research and achieve this aim. This review is of great value for guiding the research of developing botanical foods and other natural products as anti-cancer medicines from the methodological view. It’s worth being published on Cancers after revising the following points:
1. Figure 2 describes the whole development pipeline of botanical foods as anti-cancer medicine. It also highlights the outline of this whole review. Therefore, it may be good to put this figure at the beginning, but not the end, of the review to help readers catch the key points before they go into the detailed texts.
2. In session 2, the authors highlighted the importance of investigating the food-drug interaction and the complexity of phytonutrient absorption. It would be good to give some previous research works as examples to show how to systematically consider the food-drug interaction and phytonutrient absorption when designing and conducting botanical food anti-cancer studies.
3. In sessions 3.2.1 and 3.2.2, the authors recommended wisely choosing in vitro cell lines and ex vivo models when conducting preclinical studies and introduced various in vitro and ex vivo platforms. However, the suggestions are too general. The authors should discuss more detailed examples to explain how to choose the cell lines and ex vivo models to meet the specific research aims in botanical food anti-cancer studies.
4. In the Discussion and Outlook part, the authors discussed a debatable point of consuming whole-food or extracted supplements. The authors mentioned that their studies about WBM suggested the advantages of consuming whole-food in targeting different pathways simultaneously. However, it may not be a universal conclusion that whole foods are better than extracted supplements. It would be good to discuss the advantages of extracted supplements, for example, the ease of performing quality control and mechanistic study.
5. Supplement Table 1 listed many different anti-cancer foods and their mechanistic studies. It would be good to add experimental model information (for example, in vitro cell lines, ex vivo organoids, or in vivo animal models) so that readers can easily know the applied methodologies and platforms.
Author Response
We appreciate your supportive and constructive comments. Your valuable suggestions have undoubtedly helped to improve the integrity and readability of our manuscript. We have carefully considered your recommendations and revised our manuscript accordingly.
Please see the attachment.

Reviewer 3 Report
The novelty of the manuscript is limited, and the readership of this paper is limited.
Author Response
We sincerely appreciate that you have taken the time to read through our manuscript. You have mentioned that the novelty and readership of our manuscript is limited. After extensive literature research, we believe that the reviewed topics of our manuscript has been mostly overlooked, and our review provides invaluable information to readers, especially if they are interested in learning about methodologies that can aid their research with developing botanical foods as anticancer medicines. Considering valuable suggestions by the other reviewers, we have carefully revised the manuscript. We hope that the revision has made our points clearer.

Reviewer 4 Report
Dear Authors
You provide a comprehensive Review concerning Mechanism-Driven and Clinically Focused Development of Botanical Foods as Multitarget Anticancer Medicine by collecting Perspectives and Insights from Preclinical Studies, IND Applications as well as Early Phase Clinical Trials
The methodology is acceptable and objetives are achieved with clear summaries to enable scientists conceptualize and design better preclinical and clinical research to understand the potential of botanical foods as anticancer therapies .
The Preclinical Studies design is well addressed with proper consideration of the Regulatory, Scientific, and Clinical Settings Prior to Human Studies and highlights are meaningful in relation with main regulatory demands.
Also interesting and quite important is from my point of view the suggestion that patients should consume whole botanical foods with phytochemical contents that have been proven to be safe rather than supplements and concentrated extracts. I agree that it is necessary to conduct scientific pre-clinical and clinical studies under a regulatory framework to determine the potential use of whole foods in cancer treatment.
Author Response
Thank you for your supportive comments. We are grateful that you have taken the time to go through our manuscript. We have since made revisions based on the comments of other reviewers, and we welcome you to re-read our manuscript.

Round 2
Reviewer 1 Report
The authors have been very responsive to the comments of the reviewers and the paper has improved considerably. There are still a few major issues:
The inclusion of a case study on the mushroom studies of the authors in section 5 is great. However, I would suggest to move some text in the other sections to section 5: Lines 288-325, 372-379, 439-445, 453-455, 470-473, 475-481, 583-593, 615-626, 650-654, 679-686, and 711-713. Some subheadings in section 5 may be helpful in bringing more structure in this section.
Missing in Figure 2 and accompanying text is the role of the potentially critical gut microbiome although this is briefly mentioned in lines 710-711. One often overlooked issue is the possibility that the botanical or drug under study may alter the gut microbiome and thereby potentially affect their anticancer activity or other bioactive factors that are orally consumed or excreted into the gut by the liver, pancreas or gut itself.
In line 350 it is mentioned that in vitro cell models have their limitations, but these limitations are not addressed even though they are critical in evaluating the potential for translation to in vivo of findings with these models. This issue also pertains to the section regarding ex vivo models.
Still lacking is a discussion of the actual diet experimental animals are fed in in vivo model studies. Should this be an open or closed formula natural ingredient diet or a semipurified diet? Which type of diet is most appropriate for what find of botanical or food tested?
This reviewer takes some issue with the suggestion that gavage treatment is preferable over incorporation of botanicals is the diet. With careful food consumption measurements one can arrive at quite accurate estimates of delivered dose via the diet. Furthermore, diet delivery may be the only way in which large amounts of a botanical food such as a protein supplement can be administered. Gavage treatment is more appropriate for botanicals that are consumed by humans in pill or capsule form or other ways of bolus dose administration.
Also, here are some suggestions to modify some sentences to improve clarity:
Lines 64-65:
Our review of these six databases has led to compilation identification of the ten botanical foods that have evidence-based anticancer activity (Table 3).
Lines 197-199:
… how botanical food products (whole foods, food extracts, and mixtures taken as dietary supplements) are liberated via oral consumption and are subsequently digested in digestive organs after oral consumption.
Lines 210-211:
Botanical food products that contain orally bioavailable chemicals that have or are suspected to have anticancer properties are candidates 210 for in-depth investigations.
Line 707:
…. intervention studies, it is occasionally often impractical or impossible to design ….
Line 474:
…. which involves giving a group of animals the an approved drug that reflects ….
Author Response
Comments and Suggestions for Authors
The authors have been very responsive to the comments of the reviewers and the paper has improved considerably. There are still a few major issues:
Authors: We sincerely appreciate that you have taken the time to read through our manuscript in details. Your valuable suggestions have undoubtedly helped to improve our manuscript. We have carefully taken your recommendations and revised our manuscript accordingly.
The inclusion of a case study on the mushroom studies of the authors in section 5 is great. However, I would suggest moving some text in the other sections to section 5: Lines 288-325, 372-379, 439-445, 453-455, 470-473, 475-481, 583-593, 615-626, 650-654, 679-686, and 711-713. Some subheadings in section 5 may be helpful in bringing more structure in this section.
Authors: Accordingly, we have incorporated the above-mentioned text into Section 5. This section is reorganized by providing subheadings in the logical order which is compiled with the “stepwise development pipeline” we have proposed as Figure 1. The sub-sectional titles listed as “5.1 Prior-human experience of mushroom products as anticancer medicine”, “5.2 Clinical-relevant in vitro bioassay established in our lab”, “5.3 Various preclinical models and multi-targets profiling approaches applied in our study”, “5.4 Totality-of-the-evidence approach for WBM product as IND”, “5.5 Designed and conducted clinical interventional studies in human”. Those changes are made to Section 5, lines 644 to 821.
Missing in Figure 2 and accompanying text is the role of the potentially critical gut microbiome although this is briefly mentioned in lines 710-711. One often overlooked issue is the possibility that the botanical or drug under study may alter the gut microbiome and thereby potentially affect their anticancer activity or other bioactive factors that are orally consumed or excreted into the gut by the liver, pancreas or gut itself.
Authors: We acknowledged the critical perspectives on the gut microbiome. Indeed, besides the gut epithelial barrier and mucosal immune system, gut-microbiome (including microbiome-encoded enzymes) is proven as another determinant of drug pharmacokinetics and accordingly therapeutic response by modifying the drugs’ physicochemical properties (i.e., solubility and permeability) and/or transforming the drugs’ activity (i.e., microbe-mediated prodrug activation, or drug metabolism or inactivation). Particularly for botanical food products which are orally consumed, extensive research has suggested a mutual interface between gut-microbiome and botanical food products. Phytochemicals present in botanical foods can modulate the gut microbiota composition by selectively inhibiting some pathogenic microorganisms thus reducing competition within microbial populations, and ultimately promoting homeostasis within the gastrointestinal tract. Furthermore, the gut microbiome-encoded enzymes metabolize phytochemicals, in turn, may also increase their bioavailability and bioactivity. In brief, inter-individual variability in gut-microbiome composition would influence individual response to botanical foods’ anticancer response. Therefore, the researchers must not overlook the impacts on/from gut-microbiome in botanical food intervention studies. Those discussions were added to Section 2, line 247 to 261. Figure 2 is also modified accordingly.
In line 350 it is mentioned that in vitro cell models have their limitations, but these limitations are not addressed even though they are critical in evaluating the potential for translation to in vivo of findings with these models. This issue also pertains to the section regarding ex vivo models.
Authors: We appreciated the comments on those points. We thus addressed the points according in the text. Those changes were made to Section 3.2.1, line 333 to 341, and Section 3.2.2, line 365 to 383.
Still lacking is a discussion of the actual diet experimental animals are fed in in vivo model studies. Should this be an open or closed formula natural ingredient diet or a semi-purified diet? Which type of diet is most appropriate for what find of botanical or food tested? This reviewer takes some issue with the suggestion that gavage treatment is preferable over incorporation of botanicals is the diet. With careful food consumption measurements one can arrive at quite accurate estimates of delivered dose via the diet. Furthermore, diet delivery may be the only way in which large amounts of a botanical food such as a protein supplement can be administered. Gavage treatment is more appropriate for botanicals that are consumed by humans in pill or capsule form or other ways of bolus dose administration.
Authors: We acknowledged that the reviewer agreed with our suggestions on gavage treatment in animal studies. Regarding to the questions and concerns on experimental diet applied to the animals, according to the literature and our experiences, there is still a lack of standardization in diet design in food intervention animal studies of cancer. The majority of food intervention studies administer the foods via the diet, either by incorporation of experimental food ingredients into an open formula semi-purified diet, or the addition of a food extract to a standard chow diet. However, many closed-formula natural ingredient laboratory chow diets contain unknown compounds that may impact the study endpoints and results. For example, soy is commonly used as a protein source in laboratory chows, and this may inadvertently introduce phytoestrogens into the diet in the form of isoflavones. Open-formula semi-purified diets provide a benefit over standard chow if the food intervention involves a modification of lipids (e.g., corn oil, nuts oil). For foods with defined active constituents, custom manufacturing active ingredients into semi-purified diets or into standard chow diets is possible.
Some studies have adopted oral gavage, where tubes are used to deliver an exact amount of the food crude extract to the animal to ensure that the food products are ingested regardless of regular and background diet intake. For the experimental foods with unknown active ingredients, administration of food extract via oral gavage should be considered. Furthermore, in humans, botanical foods may only be delivered through diet in large amounts. Generally, oral administration is more appropriate when botanicals are consumed as pills, or capsules, other than bolus doses by humans. Those discussions were added to Section 3.2.3, line 424 to 449.
Also, here are some suggestions to modify some sentences to improve clarity:
Lines 64-65:
Our review of these six databases has led to compilation identification of the ten botanical foods that have evidence-based anticancer activity (Table 3).
Lines 197-199:
how botanical food products (whole foods, food extracts, and mixtures taken as dietary supplements) are liberated via oral consumption and are subsequently digested in digestive organs after oral consumption.
Lines 210-211:
Botanical food products that contain orally bioavailable chemicals that have or are suspected to have anticancer properties are candidates 210 for in-depth investigations.
Line 707:
…. intervention studies, it is occasionally often impractical or impossible to design ….
Line 474:
…. which involves giving a group of animals an approved drug that reflects ….
Authors: Thanks for helping us to improve the clarity. Those points have been revised accordingly in the manuscript.
Reviewer 3 Report
In this review, with the consideration of the regulatory framework of the USFDA, authors share the collective experiences and lessons learned from 20 years of defining anti-cancer foods, focusing on the critical aspects of preclinical studies that are required for IND application, as well as the checkpoints needed for early phase clinical trials. This manuscript can be accepted after minor revision, and the reviewer would like to review the revision again. Apart from grammar editing, several suggestions were given on this manuscript.
1, More botanical food products with cancer-fighting potential should be listed, such as tea, Dioscorea species. These reports can be referenced: https://doi.org/10.1016/j.lwt.2015.01.003; doi: 10.1080/10408398.2017.1347556; https://doi.org/10.1007/s11101-017-9505-5; doi: 10.1186/s13065-018-0423-4; DOI: 10.1186/1471-2091-15-19.
2, The resolution of the figures should be improved, and the format of the reference list needs to be checked again.
Author Response
Comments and Suggestions for Authors
In this review, with the consideration of the regulatory framework of the USFDA, authors share the collective experiences and lessons learned from 20 years of defining anti-cancer foods, focusing on the critical aspects of preclinical studies that are required for IND application, as well as the checkpoints needed for early phase clinical trials. This manuscript can be accepted after minor revision, and the reviewer would like to review the revision again. Apart from grammar editing, several suggestions were given on this manuscript.
Authors: We appreciate the reviewer’s supportive comments. We have carefully taken your recommendations and revised our manuscript accordingly.
1, More botanical food products with cancer-fighting potential should be listed, such as tea, Dioscorea species. These reports can be referenced: https://doi.org/10.1016/j.lwt.2015.01.003; doi: 10.1080/10408398.2017.1347556; https://doi.org/10.1007/s11101-017-9505-5; doi: 10.1186/s13065-018-0423-4; DOI: 10.1186/1471-2091-15-19.
Authors: We acknowledged the suggestion that more botanical products with anticancer potentials should be listed. However, due to the focus of this manuscript, we merely provide a few of them as examples. The intent of this review is not to provide an overview of anticancer botanicals along with their phytochemical profile and mechanism of action, because there have been a wide variety of review articles in this direction. Rather, we would like to share our collective experiences in the characterization of anticancer foods under USFDA regulatory framework. We agreed that food list in current manuscript is not comprehensive, we thus addressed in the main text that “Aside from the foods listed in Table 3, there are many other botanical products that are also studied for their anticancer activities, such as tea, coffee, spinach, etc. The six databases listed in Table 2 provide comprehensive information about these botanicals”. Those discussions were added to Section 2, line 169 to 172.
2, The resolution of the figures should be improved, and the format of the reference list needs to be checked again.
Authors: The changes were made to Figures and the format of the reference list were revised.